# Anti-Apoptotic Effects of Recombinant Human Hepatocyte Growth Factor on Hepatocytes Were Associated with Intrahepatic Hemorrhage Suppression Indicated by the Preservation of Prothrombin Time

**DOI:** 10.3390/ijms20081821

**Published:** 2019-04-12

**Authors:** Sotaro Motoi, Hiroko Toyoda, Takashi Obara, Etsuko Ohta, Yoshihisa Arita, Kana Negishi, Katsuhiro Moriya, Yoshikazu Kuboi, Motohiro Soejima, Toshio Imai, Akio Ido, Hirohito Tsubouchi, Tetsu Kawano

**Affiliations:** 1Eisai Co., Ltd., KAN Product Creation Unit, Eisai Product Creation Systems, 5-1-3 Tokodai, Tsukuba, Ibaraki 3002635, Japan; s-motoi@kan.eisai.co.jp (S.M.); htoyoda1106@gmail.com (H.T.); t-obara@hhc.eisai.co.jp (T.O.); e2-ota@hhc.eisai.co.jp (E.O.); moriya@hhc.eisai.co.jp (K.M.); y-kuboi@kan.eisai.co.jp (Y.K.); m-soejima@kan.eisai.co.jp (M.S.); 2KAN Research Institute, Inc., 6-8-2 Minatojima-Minamimachi, Chuo-Ku, Kobe, Hyogo 6500047, Japan; y-arita@kan.eisai.co.jp (Y.A.); k-negishi@kan.eisai.co.jp (K.N.); t-imai@kan.eisai.co.jp (T.I.); 3Digestive and Lifestyle Diseases, Department of Human and Environmental Sciences, Kagoshima University Graduate School of Medical and Dental Sciences, 8-35-1 Sakuragaoka, Kagoshima 8908544, Japan; ido-akio@m2.kufm.kagoshima-u.ac.jp; 4Department of HGF Tissue Repair and Regenerative Medicine, Kagoshima University Graduate School of Medical and Dental Sciences, 8-35-1 Sakuragaoka, Kagoshima 8908544, Japan; 5Department of Gastroenterology and Hepatology, Kagoshima City Hospital, 37-1 Uearata-cho, Kagoshima 8908760, Japan; tsubouchi-h62@kch.kagoshima.jp

**Keywords:** acute liver failure, cytokeratin-18, Fas antigen, hepatocyte growth factor, prothrombin time

## Abstract

Hepatocyte growth factor (HGF) is an endogenously expressed bioactive substance that has a strong anti-apoptotic effect. In this study, we biochemically and histologically characterized the effects of rh-HGF on in vitro human hepatocyte injury and mouse acute liver failure (ALF) models, both of which were induced by antibody-mediated Fas signaling. rh-HGF inhibited intracellular caspase-3/7 activation and cytokeratin 18 (CK-18) fragment release in both models. Histologically, rh-HGF dramatically suppressed parenchymal damage and intrahepatic hemorrhage. Among the laboratory parameters, prothrombin time (PT) was strongly preserved by rh-HGF, and PT was well correlated with the degree of intrahepatic hemorrhage. These results showed that the anti-apoptotic effect of rh-HGF on hepatocytes coincided strikingly with the suppression of intrahepatic hemorrhage. PT was considered to be the best parameter that correlated with the intrahepatic hemorrhages associated with hepatocellular damage. The action of rh-HGF might derive not only from its anti-apoptosis effects on liver parenchymal cells but also from its stabilization of structural and vasculature integrity.

## 1. Introduction

Human hepatocyte growth factor (HGF) is an endogenously expressed bioactive substance that was originally discovered in the serum of fulminant hepatitis patients because of its powerful ability to induce hepatocyte proliferation; thus, it was originally recognized as a hepatocyte growth factor [1,2,3]. Since its discovery, HGF has been found to induce strong and rapid anti-apoptotic effects against liver injury in mice [4,5]. These characteristics suggest the clinical utility of HGF for treating acute liver injury, such as acute liver failure (ALF). ALF is a devastating clinical syndrome with high mortality [6,7]. The immediate initiation of intensive medical support is required, but liver transplantation is the only treatment that considerably improves prognosis [8]. In clinical settings, prothrombin time (PT) is one of the key parameters for predicting patient prognosis, and the restoration of a prolonged PT-international normalized ratio (INR) to below 1.3 predicts an increased survival rate with conservative treatments [9]. 

With the support of the Japan Agency for Medical Research and Development, the pharmaceutical development of recombinant human HGF (rh-HGF) for ALF was initiated in 2011, and this compound is currently being used in a phase I study in healthy volunteers (ClinicalTrials.gov Identifier: NCT03014895). Prior to clinical studies in actual ALF patients, we investigated the pre-clinical effects of rh-HGF in vitro and in vivo. In both of these models, Fas signaling was induced directly by anti-Fas antibodies to identify any particular features of rh-HGF treatment. In the analyses, correlations between biochemical parameters and changes in histopathological characteristics were explored. Clinically useful laboratory parameters were also evaluated for their correlation with histological characteristics. In particular, we included PT as a parameter because it is important in the clinical treatment of ALF.

## 2. Results

### 2.1. Recombinant Human Hepatocyte Growth Factor (rh-HGF) Inhibited Fas-Induced Apoptosis in Primary Cultured Human Hepatocytes

The anti-apoptotic effect of rh-HGF was evaluated in cultured human hepatocytes. Primary cultured human hepatocytes were exposed to anti-Fas antibodies to induce apoptotic cell injury. The anti-Fas antibody drastically increased intracellular caspase-3/7 activity, while the control antibody showed no activity (Figure 1A). Cell apoptosis coincided with the increase of cytokeratin 18 (CK-18) M30 fragment concentration in the supernatant that resulted from its release from injured cells (Figure 1B). Intracellular caspase-3/7 activity was suppressed by rh-HGF in a dose-dependent manner, and the mean of the half-maximal inhibitory concentration (IC_50_) value of three batches was calculated as 0.82 ng/mL (95% CI: 0.0095–71). The release of CK-18 M30 fragment into the culture medium was inhibited at a mean IC_50_ value of 0.39 ng/mL (95% CI: 0.013–11). These results demonstrated that rh-HGF suppressed human hepatocyte injury that was directly induced by Fas-mediated death signaling.

### 2.2. rh-HGF Suppressed Mouse Acute Liver Failure (ALF) via Anti-Apoptotic Effects and Preserved Prothrombin Time (PT)

The in vivo effects of rh-HGF were evaluated using an anti-Fas antibody (Jo2)-induced mouse ALF model. We investigated clinical parameters such as serum aspartate transaminase (AST), alanine transaminase (ALT), and blood PT. Consistent with our hypothesis, increases in serum AST and ALT levels were significantly inhibited by rh-HGF at doses of 1.5 mg/kg (*p* = 0.0009, 0.0014, respectively) and 5 mg/kg (*p* = 0.0017, 0.0015, respectively); however, the average values still exceeded 4000 U/L for AST and 10,000 U/L for ALT (Figure 2A,B). On the other hand, the effect against PT prolongation was significant because rh-HGF treatment resulted in PT values close to the normal values (Figure 2C, *p* < 0.0001 in all groups treated with rh-HGF). Parenchymal caspase activity and CK-18 fragment serum levels, both of which might directly reflect liver cell apoptosis, were also determined. Diffuse intracellular caspase activation (Figure 2D, Control) and extracellular release of CK-18 fragments (Figure 2E, Control) were reduced by rh-HGF (Figure 2D,E, rh-HGF).

### 2.3. rh-HGF Treatment Reduced Hepatocellular Damage and Prevented Diffuse Hemorrhage in the Liver

To determine the changes in histopathological characteristics induced by rh-HGF treatment, microscopic images were scored according to the degree of hepatocellular damage, hemorrhage, and immunohistochemically evaluated intracellular caspase activity. Severity of hepatocellular damage with a disorganized liver structure clearly decreased (Figure 3A,B), and caspase induction was reduced (Figure 3D) by rh-HGF. In the animals treated with rh-HGF, intrahepatic hemorrhages were not observed, although there were scattered areas of damaged hepatocytes (Figure 3A,C).

### 2.4. Intrahepatic Hemorrhage Suppression was Well Correlated with PT Preservation

We investigated which blood parameters could reflect the prevention of hemorrhage by rh-HGF. AST, ALT, and PT were substantially correlated with hemorrhage scores as their contribution values (R^2^) were 0.8784, 0.8195, and 0.9014, respectively (Figure 4A–C, left panels). Among the rh-HGF-treated mice, the value dispersion was the narrowest for PT (Figure 4A–C, right panels), which meant that strong preservation of PT was the best parameter reflecting the hemorrhage suppression effects of rh-HGF.

## 3. Discussion

The prognosis of ALF has been improved by etiology-based treatments and advanced artificial liver support, but there is no doubt that further improvements are required. Especially for patients with hepatic coma, liver transplantation remains the only curative measure. Because transplantation is not always available, a new treatment that can stop the progression of liver damage is needed. HGF is thought to be a potent treatment because of its strong anti-apoptotic effect and stimulation of tissue repair activities [4,5,10]. Although a clinical trial of rh-HGF was previously conducted (Clinicaltrial.gov identifier: NCT03014895), its therapeutic efficacy in ALF remains unknown. Additionally, histological assessments of the pharmacological effects of rh-HGF are difficult in clinical settings. Therefore, in the present study, we first confirmed the anti-apoptotic effect of rh-HGF using cultured human hepatocytes; thereafter, we investigated the histological characteristics of the liver after rh-HGF treatment in an ALF mouse model. We included mouse PT measurements because PT is one of the most important parameters for assessing the functional liver mass in ALF. 

In the present study, anti-apoptotic effects of rh-HGF were clearly observed in a human hepatocyte injury model (Figure 1) and mouse ALF model (Figure 2). In both models, rh-HGF suppressed cell death machinery (Figure 1A and Figure 2D), which was associated with the prevention of the release of cell components to outside of cells (Figure 1B and Figure 2A,B,E). The in vivo effects of rh-HGF were well depicted by the strong suppression of hepatocellular damage (Figure 3A,B) and an associated decrease in intracellular caspase activation (Figure 3D). Surprisingly, the occurrence of intrahepatic hemorrhages was dramatically reduced to non-detectable levels with rh-HGF (Figure 3A,C); however, there were scattered areas of damaged hepatocytes with increased AST and ALT levels (Figure 2A,B and Figure 3A). Diffuse and massive intrahepatic hemorrhage is known to be one of the histological features of ALF, and the complete suppression of hemorrhage occurrence is an outstandingly beneficial feature of rh-HGF treatment.

Histological evaluation of the damaged liver is not always conducted in ALF patients, and there are no known clinical parameters with which we can determine the effect of rh-HGF against intrahepatic hemorrhage. Among the clinically conventional parameters, PT was found to be the best parameter that strongly correlated with the decrease of hemorrhages in our study (Figure 4). In this sense, PT may reflect the severity of intrahepatic vascular damage as well as provide useful information regarding residual functional liver mass.

HGF is known to promote endothelial cell proliferation and angiogenesis by binding c-Met on vascular endothelial cells [11], and it is also reported to protect advanced glycation end products (AGE)/reactive oxygen species (ROS)-mediated endothelial apoptosis via phosphatidylinositol 3-kinase (PI3K)/ protein kinase B (Akt) activation [12]. In light of these facts, the findings obtained in this study raise a beneficial possibility that rh-HGF might stabilize microvasculature integrity even under conditions of parenchymal damage in the liver due to ALF. In future clinical studies in ALF patients, the effects on microvascular injury should be evaluated using blood vessel biomarkers, which may provide further clues for understanding the effects of rh-HGF on the vasculature and, of course, help to determine an accurate dosing regimen for rh-HGF.

In conclusion, the study results of this report should support an idea that rh-HGF might significantly benefit ALF patients for whom no treatment is currently available by directly inhibiting liver parenchymal damage. In addition, the effect of rh-HGF might be derived not only from its anti-apoptotic effects on parenchymal cells but also from stabilization of microvasculature integrity. Although further analysis is required to determine the dominant action of HGF, at least in the clinical setting, PT might provide a clue for predicting the histological consequences of rh-HGF treatment, which will be associated with the amelioration of hepatic structure destruction and hemorrhages.

## 4. Materials and Methods

### 4.1. Recombinant Human HGF

Recombinant human HGF (rh-HGF) was manufactured by Eisai Co. Ltd. (Tokyo, Japan), and its designated code name was E3112. Briefly, human HGF was firstly expressed as pro-HGF by using a mammalian cell line, thereafter, it was cleaved by recombinant hepatocyte growth factor activator (pro-HGFA) that was also produced by a mammalian cell line. All culturing was conducted with serum-free culture media. E3112 manufacturing proceeded according to the guidelines of good manufacturing practices (GMPs).

### 4.2. In vitro Human Hepatocyte Injury Model

Cryopreserved human hepatocytes (Biopredic International, Rennes, France) were thawed and seeded at 40,000 cells/well in collagen I-coated 96-well plates. Then, the cells were pre-incubated with or without rh-HGF for 6 h at final concentrations of 0, 0.1, 0.3, 1, 3, 10, 30, 100, 300, 1000, or 3000 ng/mL at 37 °C under a 5% CO2 atmosphere. Thereafter, 60 ng/mL anti-human Fas monoclonal antibody (2R2, Enzo Life Sciences, Farmingdale, NY, USA) or mouse IgG3 isotype control was added and incubated for 24 h. After incubation, intracellular caspase-3/7 activity was measured using a Caspase-Glo 3/7 Assay Kit (Promega, Madison, WI, USA).

Caspase-3/7 activity relative to that of the control (fold of control) was determined using the following formula:

Fold of control = (relative luminescent unit (RLU) of the test condition − mean background RLU)/(mean RLU of the no antibody − mean background RLU).
(1)


Soluble human CK-18 M30 levels in the culture supernatants were measured by using an M30 Apoptosense^®®^ ELISA Kit (Peviva, Bromma, Sweden). 

### 4.3. Mouse ALF Model

Six-week-old male BALB/c mice (Charles River Japan, Kanagawa, Japan) received intravenous injections of 0.23 mg/kg anti-Fas monoclonal antibody (Jo2; BD Pharmingen, San Diego, CA, USA). One hour before the anti-Fas antibody injection, rh-HGF was injected intravenously at doses of 0.5, 1.5, 5, or 15 mg/kg. PBS was injected as a control. Five hours after anti-Fas antibody administration, PT was evaluated using a Coaguchek XS system (Roche Diagnostics, Basel, Switzerland). Thereafter, mice were sacrificed to collect blood with which serum AST and ALT levels were measured by a Clinical Analyzer 7180 (Hitachi High-Technologies Corporation, Tokyo, Japan) and diagnostic reagent kits for each marker (Wako Pure Chemical Industries Ltd., Osaka, Japan). In another experiment, 1.5 mg/kg rh-HGF was injected following the same design described above. Serum was collected at 4 h after the anti-Fas injection for CK-18 analysis by Western blotting, and the livers were collected at 4 or 5 h after the injection for histological analyses. PT value of normal mouse was developed by using five normal mice without any treatments. All protocols were approved by the Institutional Animal Care and Use Committee and were performed according to Eisai Animal Experimentation Regulations (approval number: 15-D-0139-001, approval date: 29 September 2015).

### 4.4. Histopathological Analysis and Immunohistochemistry

Livers were fixed in 10% formalin and embedded in paraffin. Four-micrometer sections were prepared and stained with hematoxylin-eosin (HE) for histopathological analysis. For the immunohistochemical detection of intracellular cleaved caspase-3, sections were stained with an anti-cleaved caspase-3 antibody (Cell Signaling Technology, Beverly, MA, USA) according to the manufacturer’s instructions. Quantification of the cleaved caspase-3-positive area was calculated by Image J software, version 1.51h (National Institutes of Health, Bethesda, MD, USA).

### 4.5. Western Blot Analysis of Serum CK-18 in Mice

Serum CK-18 was purified by immunoprecipitation using an anti-CK-18 polyclonal antibody (Abcam, Cambridge, UK) and then analyzed using SDS-PAGE for standard Western blotting. The CK-18 protein was detected using an anti-CK-18 antibody (Abcam), peroxidase-conjugated anti-rabbit secondary antibodies (Cell signaling technology, Beverly, MA, USA), and enhanced chemiluminescence (ECL) plus Western blotting detection reagent (GE Healthcare, Chicago, IL, USA).

### 4.6. Statistics

The data were presented as the mean ± standard error of the mean (SEM). Statistical analyses were performed using GraphPad Prism version 7.02 (GraphPad Software, La Jolla, CA, USA). *p* values <0.05 were considered statistically significant.

## Figures and Tables

**Figure 1 ijms-20-01821-f001:**
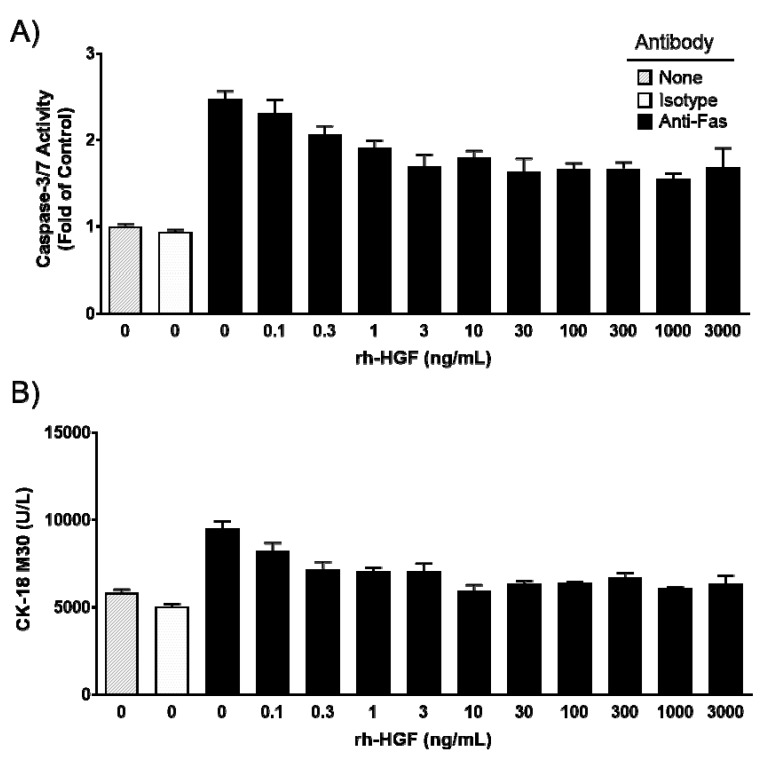
Effects of recombinant human hepatocyte growth factor (rh-HGF) in a Fas antibody-induced human hepatocyte injury model. (**A**) Caspase-3/7 activity, (**B**) CK-18 M30 fragment concentration. Each bar represents the mean ± SEM of representative hepatocytes cultured in triplicate. None, no antibody; Isotype, mouse Immunoglobulin G3 (IgG3) isotype control antibody; anti-Fas, anti-human Fas monoclonal antibody.

**Figure 2 ijms-20-01821-f002:**
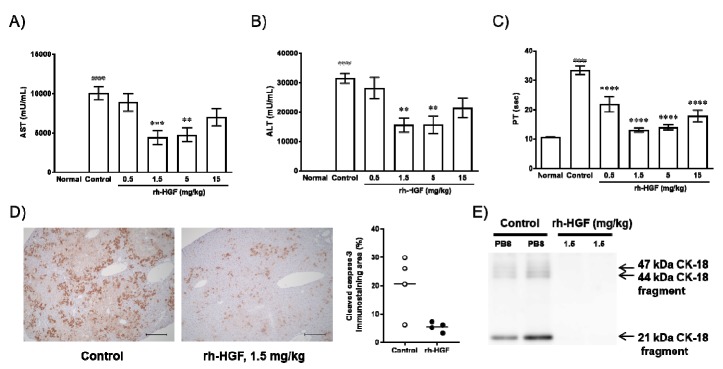
Effects of rh-HGF in an anti-Fas antibody-induced mouse acute liver failure (ALF) model. (**A**) Aspartame transaminase (AST), (**B**) alanine transaminase (ALT), and (**C**) prothrombin time (PT). #### *p* < 0.0001: comparison between the normal and control groups (unpaired t-test). Phosphate-buffered saline (PBS) was injected in the control group. ** *p* < 0.01, *** *p* < 0.001, and **** *p* < 0.0001: comparison between the control and rh-HGF-treated groups (Dunnett’s multiple comparison test). The data are presented as the mean ± SEM (*n* = 5–8). (**D**) Cleaved caspase-3 immunostaining. The images are representative of four animals from each group. Quantification of cleaved caspase-3 immunostaining area (%) was calculated by using Image J software (right panel). Scale bars, 200 µm. (**E**) Serum CK-18 levels. CK-18 levels were shown by Western blotting. Each lane shows individual animals.

**Figure 3 ijms-20-01821-f003:**
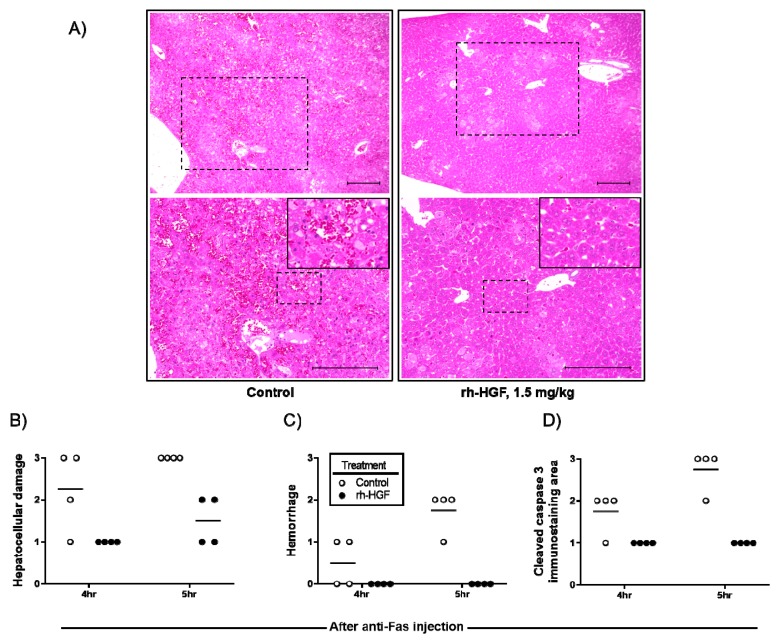
Histological analysis of anti-Fas antibody-induced mouse ALF livers. (**A**) Representative images of the livers from PBS (control, left rectangle) or rh-HGF (right rectangle) treated mice are shown at low (upper panels) or high magnification (lower panels) with hematoxylin-eosin (HE) staining. Dotted rectangles in upper panels were magnified into lower panels. Further magnified images (insets in the lower panels) were derived from the rectangle in lower panels to show representative lesion of intrahepatic hemorrhage and to compare its severity with rh-HGF treated mice. Scale bars, 200 µm. (**B**) The degree of hepatocellular damage was scored qualitatively as unremarkable, slight, moderate, or marked, noted as 0, 1, 2, or 3, respectively. (**C**) The degree of hemorrhage was also scored qualitatively as no hemorrhage, slight hemorrhage, moderate hemorrhage, or marked hemorrhage as 0, 1, 2, or 3, respectively. (**D**) The degree of cleaved caspase-3 immunostaining area (%) was classified as 0%, 0–10%, 10–30%, or over 30% as 0, 1, 2, or 3, respectively. Control (PBS, open circles) or rh-HGF (1.5 mg/kg, closed circles) was administered to the mice. hr: hour.

**Figure 4 ijms-20-01821-f004:**
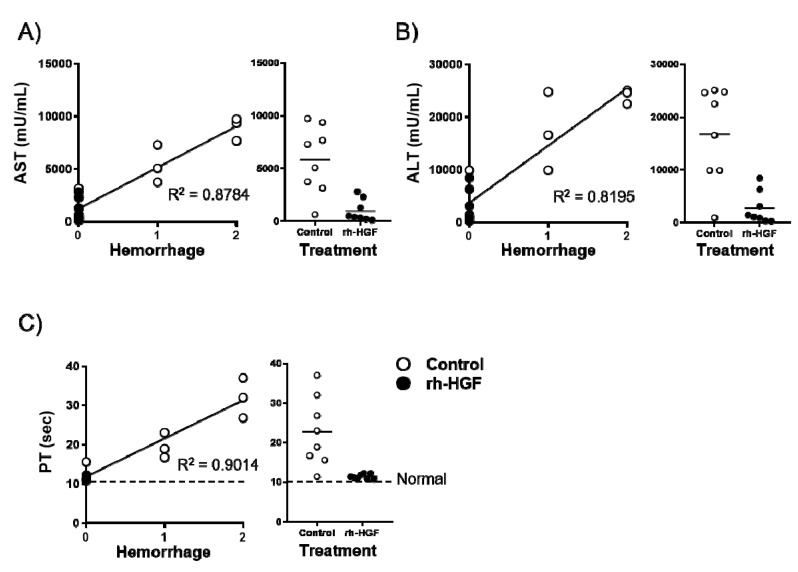
Correlation between the degree of hemorrhage and blood parameters. Correlation analysis between the degree of hemorrhage and (**A**) AST, (**B**) ALT, and (**C**) PT were conducted, and their resulting contribution values (R^2^) were shown in the left panels. Distributions of each blood parameter were presented according to the treatment (right panels). Average PT (10.6 s) in normal mice was indicated with a dotted line in (**C**). Control (PBS, open circles) or rh-HGF (1.5 mg/kg, closed circles) was administered to the mice.

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
