# Peer review of "Anti-Apoptotic Effects of Recombinant Human Hepatocyte Growth Factor on Hepatocytes Were Associated with Intrahepatic Hemorrhage Suppression Indicated by the Preservation of Prothrombin Time"

_ijms, 2019, doi:10.3390/ijms20081821_

Reviewer 1 Report

In this manuscript, Motoi and coworkers characterized the strong and rapid anti-apoptotic effect induced by rh-HGF against liver injury in mice in vivo. Culture human hepatocytes were also used to evaluate the apoptotic cell injury caused by the anti-Fas antibody. Overall, I recommend acceptance of this work after addressing the following issues.

1.       Except for quantification of the caspase-3/7 activity and CK-18 M30 concentration, more direct demonstration such as immunofluorescence images are important to support the alleviated hepatocyte injury.

2.       Is there any evidence from the literature that PT was the best parameter reflecting the hemorrhage suppression effect of rh-HGF?

3.       Any side effect of rh-HGF clinically?

4.       What is the cellular mechanism of the rh-HGF against apoptotic injury?

Author Response

Response to Reviewer 1:

The reviewer’s Comments and Suggestions for Authors are shown in Times New Roman italics. Our responses are shown in Arial and blue font.

We thank the reviewer for their encouraging, helpful and detailed comments regarding our work. We have revised the manuscript according to your suggestions, which have greatly contributed to improving our paper. With this added information, we really hope that our research makes a valuable contribution to understand the pharmacological effects of rh-HGF as a new medicine for ALF patients.

Please find below our point-by-point responses to the reviewer’s comments:

1.          We are very grateful to the reviewer’s important comment. As the reviewer pointed out, we also think it is very important to show direct demonstration of hepatocyte apoptosis in the model. Unfortunately, we have not yet tested immunofluorescent staining such as for cleaved caspases and TUNEL–staining [1], however, we repeatedly observed the morphological changes of hepatocytes treated with anti-Fas antibody with bizarre shaping of nuclei and shrinkage of cytoplasm, and rh-HGF could prevent those morphological changes. These observations are similar to the previous work demonstrating the effect of HGF on Fas-mediated apoptosis in mouse primary cultured hepatocytes [2]. In this manuscript, we cannot show those immunofluorescent staining images until the allowed deadline for the revision, however we surely recognized the importance. Therefore, we would like to consider these immunostaining for future study.

2.      We appreciate the reviewer’s important suggestion. To our knowledge, there is no evidence that PT was the best parameter reflecting the hemorrhage suppression effect of rh-HGF. In a previous work, it was reported that HGF administration ameliorated parenchymal hemorrhage as well as suppression of AST and ALT elevation, however, PT prolongation was not examined [2]. Present study is the first report to show rh-HGF’s effect with PT in conjunction with histology in ALF model.

3.      As the reviewer pointed out, information of side effect of rh-HGF in clinical is quite important. Unfortunately, we don’t have any data publically available because the study report of the clinical trial with E3112 has not been published yet.

4.      We are grateful to the reviewer for raising such an important question. In the course of Fas mediated apoptotic signal, Fas-associated with death domain (FADD) degrades FLICE Inhibitory Protein (FLIP), which results in the activation of Caspase-8 that subsequently activates other death effectors such as Caspase-3 [1]. It is reported that HGF inhibited the degradation of FLIPL (the long splice form of FLIP) by activation of Akt-PI3K through Met signaling by which death signaling is halted.

1.          Moumen, A.; Ieraci, A.; Patane, S.; Sole, C.; Comella, J. X.; Dono, R.; Maina, F., Met signals hepatocyte survival by preventing Fas-triggered FLIP degradation in a PI3k-Akt-dependent manner. Hepatology 2007, 45, (5), 1210-7.

2.          Kosai, K.; Matsumoto, K.; Nagata, S.; Tsujimoto, Y.; Nakamura, T., Abrogation of Fas-induced fulminant hepatic failure in mice by hepatocyte growth factor. Biochem Biophys Res Commun 1998, 244, (3), 683-90.

Reviewer 2 Report

In this study, the authors demonstratedthat (1) HGF suppressed apoptosis of human hepatocytes in culture, (2) administration of recombinant HGF in mice model of acute liver failure caused by anti-Fas antibody suppressed apoptosis of hepatocytes as evaluated by serum aspartic acid transaminase alanine transaminase, prothrombin time, caspase-3 activation, and the release of CK-18 fragment, (3) administration of recombinant HGF in mice model of acute liver failure dramatically suppressed parenchymal damage and intrahepatic hemorrhage, and (4) the suppression of dysfunction of blood coagulation as evaluated by prothrombin time correlated with the degree of intrahepatic hemorrhage.

Suppression of acute liver failure by administration of HGF in different models have been reported. The authors obtained similar finding, while they provide new finding. Several points should be addressed.

Comments:

1.    Discussion section: Many descriptions more than half are repeat of the results. The authors should reconstruct this section to decrease the redundancy.

2.    Figure 2A~C: Individual P values should be described, not like ‘P<0.01’.< span="">

3.    Figure 2D: The caspase-3-positive area should be quantitated by image analysis. 

4.    Figure 3: The legend for Figure 3 is missing, and it should be described.

5.    Figure 3A: The histology for hemorrhage is not clear. It should be indicated by images at higher magnification or resolution.

6.    Figure 4C: Prothrombin time in normal mice should be indicated.

Author Response

Response to Reviewer 2:

The reviewer’s Comments and Suggestions for Authors are shown in Times New Roman italics. Our responses are shown in Arial and blue font.

We thank the reviewer for their encouraging, helpful and detailed comments regarding our work. We have revised the manuscript according to your suggestions, which have greatly contributed to improving our paper. With this added information, we really hope that our research makes a valuable contribution to understand the pharmacological effects of rh-HGF as a new medicine for ALF patients.

Please find below our point-by-point responses to the reviewer’s comments:

1.      We apologize for the inappropriate descriptions in discussion section. We reconstituted the discussion in the revised manuscript according to the reviewer’s suggestion. During the process, we skipped the originally proposed reference No.11 to avoid any redundancy and to improve readability.

2.    We apologize for the confusing description regarding P value. We described the individual P values for Figure 2A, 2B and 2C in Result section. In case of P values that are smaller than 0.0001, GraphPad Prism software can only describe adjusted P values (We referred to the following URL; https://www.graphpad.com/support/faq/how-to-get-exact-p-values-from-multiple-comparisons-tests-that-follow-one-way-or-two-way-anova/). Therefore, we describe the P values smaller than 0.0001 as ‘P < 0.0001’ instead of numerical values in some of the comparisons.

3.    We are grateful for the reviewer’s important suggestion. To address reviewer’s suggestion, we quantified the cleaved caspase-3 positive area (before revision we just described it as “caspase-3” but precisely saying that is “cleaved caspase-3”, so that we corrected it in the revision) and the quantitative data was newly added in Figure 2D in the revised manuscript.

4.    We apologize for the insufficient description. We corrected the legend of Figure 3 in revised manuscript.

5.    We apologize for adopting the unclear images for Figure 3A. In the high-magnified images (Figure 3A, lower panels), we indicated the typical lesion of hemorrhages (dotted rectangle) that was further magnified and shown in the insets.

6.    We apologize for the lacking data of prothrombin time of normal mice. We added dotted lines indicating prothrombin time in normal mice (10.6 sec.) in Figure 4C in the revised manuscript.

Round  2

Reviewer 2 Report

The authors revised manuscript, according to comments provided for this manuscript.